# Boolean modeling of breast cancer signaling pathways uncovers mechanisms of drug synergy

Kittisak Taoma[1,2], Marasri Ruengjitchatchawalya[1,3], Monrudee Liangruksa[4]*, Teeraphan Laomettachit[1,5]*

1 Bioinformatics and Systems Biology Program, School of Bioresources and Technology, King Mongkut's University of Technology Thonburi, Bangkok, Thailand, 2 School of Information Technology, King Mongkut's University of Technology Thonburi, Bangkok, Thailand, 3 Biotechnology Program, School of Bioresources and Technology, King Mongkut's University of Technology Thonburi, Bangkok, Thailand, 4 National Nanotechnology Center, National Science and Technology Development Agency (NSTDA), Pathum Thani, Thailand, 5 Theoretical and Computational Physics Group, Center of Excellence in Theoretical and Computational Science, King Mongkut's University of Technology Thonburi, Bangkok, Thailand

* monrudee@nanotec.or.th (ML); teeraphan.lao@kmutt.ac.th (TL)

**Data Availability Statement:** All relevant data are within the manuscript and its Supporting information files. The programming script and related data required to reproduce the simulations

## Abstract

Breast cancer is one of the most common types of cancer in females. While drug combinations have shown potential in breast cancer treatments, identifying new effective drug pairs is challenging due to the vast number of possible combinations among available compounds. Efforts have been made to accelerate the process with *in silico* predictions. Here, we developed a Boolean model of signaling pathways in breast cancer. The model was tailored to represent five breast cancer cell lines by integrating information about cell-line specific mutations, gene expression, and drug treatments. The models reproduced cell-line specific protein activities and drug-response behaviors in agreement with experimental data. Next, we proposed a calculation of protein synergy scores (*PSS*s), determining the effect of drug combinations on individual proteins' activities. The *PSS*s of selected proteins were used to investigate the synergistic effects of 150 drug combinations across five cancer cell lines. The comparison of the highest single agent (HSA) synergy scores between experiments and model predictions from the MDA-MB-231 cell line achieved the highest Pearson's correlation coefficient of 0.58 with a great balance among the classification metrics (AUC = 0.74, sensitivity = 0.63, and specificity = 0.64). Finally, we clustered drug pairs into groups based on the selected *PSS*s to gain further insights into the mechanisms underlying the observed synergistic effects of drug pairs. Clustering analysis allowed us to identify distinct patterns in the protein activities that correspond to five different modes of synergy: 1) synergistic activation of FADD and BID (extrinsic apoptosis pathway), 2) synergistic inhibition of BCL2 (intrinsic apoptosis pathway), 3) synergistic inhibition of MTORC1, 4) synergistic inhibition of ESR1, and 5) synergistic inhibition of CYCLIN D. Our approach offers a mechanistic understanding of the efficacy of drug combinations and provides direction for selecting potential drug pairs worthy of further laboratory investigation.

are available from our GitHub repository (https://github.com/kittisaktaoma/BoolSyn/).

**Funding:** T.L. and M.L. acknowledge the financial support from Thailand Science Research and Innovation (TSRI) Basic Research Fund: The fiscal year 2023 under project number FRB660073/0164 (Program Sustainable Bioeconomy). K.T. acknowledges the Petchra Pra Jom Klao Ph.D. Research Scholarship (KMUTT – NSTDA) (No: 103/2563). The funders had no role in study design, data collection and analysis, decision to publish, or preparation of the manuscript.

**Competing interests:** The authors have declared that no competing interests exist.

## Introduction

Breast cancer ranks among the most prevalent forms of cancer, with the number of new cases estimated to become approximately 3.19 million worldwide in 2040 [1]. Various breast cancer treatments are currently available, such as targeted-drug therapy, hormone therapy, and chemotherapy [2]. However, the development of drug resistance hinders the effectiveness of these treatments. For example, targeted drug treatments can trigger a rapidly adaptive resistance mechanism of cancer cells by rewiring feedback loops in the protein signaling pathways [3]. In addition, chemotherapy can introduce genomic instability, leading to new resistance clones of cancer cells [4, 5].

Promisingly, drug combination therapy has been shown to mitigate drug resistance by simultaneously targeting more than one signaling pathway, effectively diminishing proliferation or inducing apoptosis [6–10]. Additionally, combined drugs with diverse mechanisms of action allow for lower individual drug doses than when used as single agents, resulting in fewer side effects while improving patient survival [11, 12].

When two drugs are combined, their effect may interact to enhance or reduce their effectiveness compared to when used separately, referred to as synergistic and antagonistic effects, respectively [13]. Several models have been proposed to quantify the degree of synergy, such as Highest Single Agent (HSA), Loewe, ZIP, and Bliss [14]. These models share a common approach to calculating the synergy score $S$ by determining the deviation of drug combination effect $y_c$ from the expected effect $y_e$, $S = y_c − y_e$. Each model uses a different hypothesis for the expected effect calculation. For example, the HSA model assumes that the expected effect ($y_e$) of drug combination is equal to the effect of either drug A ($y_A$) or drug B ($y_B$), whichever is the greater. Then, the HSA synergy score ($S_{HSA}$) is calculated as $S_{HSA} = y_c − \max(y_A, y_B)$, where $y_c$ is the effect of the drug combination. Positive and negative values of $S_{HSA}$ indicate synergistic and antagonistic effects, respectively.

Despite the availability of high-throughput assays for drug combination screening [15–18], discovering effective drug combinations remains challenging. As the number of potential targeted drugs grows, ample combinatorial space of available drugs is largely unexplored. Therefore, there is a need for computational tools to facilitate the rational selection of potential drug pairs for further investigation in the laboratory.

Many machine-learning algorithms have been proposed for drug synergy prediction in pan-cancer cell lines [19–23]. While the prediction performances were satisfactory, much was still to be learned about the underlying mechanism of the synergy between the predicted drug pairs. The 'black-box' nature of the machine learning models made it difficult to fully understand how the predictions were made. Complementary to machine learning, mechanistic modeling is an alternative approach to understanding the mechanism of cancer's drug resistance and identifying potential drug combination intervention. Boolean modeling is one of the simplest mechanistic approaches, where kinetic parameters are not required. Thus, creating a larger protein signaling network with Boolean modeling is more feasible compared to kinetic-based models. As a result, the Boolean model can include more relevant proteins, such as drug target proteins [24]. Multiple Boolean modeling frameworks have been proposed to investigate drug resistance mechanisms [25–27] and drug combination effects [28–31] in various types of cancer. However, these models did not quantify the degree of synergy, which can be quantified experimentally as a synergy score on a continuous scale (e.g., higher scores mean higher synergy). Providing such a metric is essential to drug combination therapy research [32].

In this study, we modeled signaling pathways in breast cancer with a Boolean model, which was tailored to represent five specific breast cancer cell lines by incorporating genetic mutation and gene expression profiles. The cell-line specific models were calibrated to reproduce gene

expression profiles and drug-treatment experiments. Then, the models were used to simulate a dose-response profile of 150 drug pairs retrieved from the DrugComb database [33], and the activities of representative proteins selected by a genetic algorithm were used to calculate the synergy score. Finally, the activities of the selected proteins were used to gain insight into the mechanisms of drug combinations with both synergistic and antagonistic interactions.

## Methods

Our approach shares similarities with the methodology employed by Tsirvouli et al. (2020) [28]. The study predicted drug synergy in various cancer cell lines, including one breast cancer cell line (MDA-MB-468). The present work focused on drug synergistic effects on breast cancer, extending the analysis to five breast cancer cell lines (MCF-7, T-47D, BT-549, MDA-MB-231, and MDA-MB-468). These five cell lines are commonly used in breast cancer research to represent molecular subtypes of the ER-positive (MCF-7 and T-47D) and triple-negative (BT-549, MDA-MB-231, and MDA-MB-468) cancer (for example, see [34–37]). The five cell lines are also experimentally reported in the DrugComb database with the highest numbers of tested drug pairs among other breast cancer cell lines [33].

Fig 1 summarizes the workflow in the present study with five steps. Briefly, the first step involved reconstructing a protein signaling network that comprehensively represents breast cancer proliferation pathways by integrating information from various databases, including KEGG, SIGNOR, and SignaLink **(Step 1)**. Next, the network was translated to a Boolean model **(Step 2)**. The model was further tailored into five cell-line specific Boolean models by integrating gene expression and genetic mutation profiles of each cell line. Then, each cell-line specific model was simulated to obtain steady-state protein activities under unperturbed and drug-perturbed conditions. The protein activities of each cell line under unperturbed conditions were compared to their respective gene expression data. The results of the drug perturbation simulations were compared to the observed responses of the cell lines to drug treatments, which were collected from the literature. The Boolean rules of each cell-line specific model were manually modified during this step to match model simulations with gene expression data and observed behaviors from drug-perturbed experiments **(Step 3)**. Then, the resulting Boolean models were used to simulate the effects of drug combinations on protein activities. We proposed a calculation of protein synergy scores (*PSSs*), determining the effect of drug combinations on individual proteins' activities. The *PSSs* of selected proteins were summed together and compared to the synergistic scores from the experiments **(Step 4)**. The *PSS* profiles were also used to investigate the mechanisms by which drug pairs exert their synergistic or antagonistic effects on the cell lines **(Step 5)**. Below, we describe each step in detail.

### 1. Protein signaling network reconstruction

Breast cancer-related protein signaling pathways were retrieved from the KEGG database (https://www.genome.jp/pathway/hsa05224), which contains 76 proteins and 60 interactions (accessed on 1 March 2020). Another set of 41 proteins and an additional 176 interactions from the SIGNOR2.0 (https://signor.uniroma2.it/) and 12 interactions from SignaLink (http://signalink.org/) databases (accessed on 1 January 2022) were combined. A network was used to represent the signaling pathways, where nodes represent the proteins and edges represent the interactions. Proteins forming a complex and proteins with multiple isoforms were merged into a single entity. For example, the CYCLIN_D_c node represents the CDK4-CCND1 complex, and the AKT_i node accounts for AKT1, AKT2, and AKT3. In addition, self-interactions (auto-activation) of nine ligand nodes (e.g., estrogen; ES and progesterone; PG) were included by assuming that these ligands are produced by cancer cells in an autocrine manner [38]. The

## 1. NETWORK RECONSTRUCTION

Reconstructing a signaling network of breast cancer using information from KEGG, SIGNOR, and SignaLink.

## 2. GENERIC BOOLEAN MODEL

Converting the interactions in the network to Boolean rules.

## 3. CELL-LINE SPECIFIC MODELS

Modifying Boolean rules to represent five breast cancer cell lines. Validating the models with gene expression profiles and drug-response behaviors from the literature.

## 4. DRUG COMBINATION SIMULATIONS

Perturbing cell-line specific models with effects from drug pairs from the DrugComb database. Calculating the synergy score.

## 5. MECHANISM EXPLANATIONS

Investigating the mechanisms of drug pairs from the drug-response protein activities.

**Fig 1. The workflow of the present study.**

resulting protein signaling network comprises 117 nodes and 257 edges (Fig 2). The complete node and interaction list from KEGG, SIGNOR2.0, and SignaLink can be found in S1 File.

## 2. Generic Boolean model

We implemented the Boolean modeling approach to represent the network, where each node was associated with a value of either 0 or 1 to represent the protein activity/level (0 = OFF/low

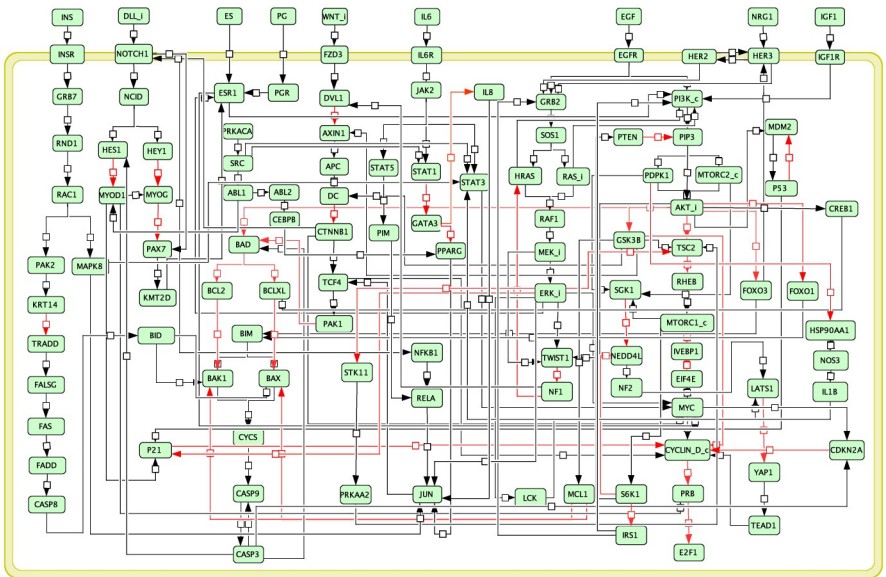

**Fig 2. A generic protein signaling network of breast cancer pathways.** The network was reconstructed using information from KEGG, SIGNOR, and SignaLink databases. Nodes represent protein entities and edges represent the interactions between them. Proteins forming a complex were merged into a single node (names ending with '_c', e.g., CYCLIN_D_c represents the CDK4-CCND1 complex). Proteins with multiple isoforms were also merged into a single node (names ending with '_i', e.g., AKT_i represents AKT1, AKT2, and AKT3). The black and red arrows represent activating and inhibitory regulation, respectively. The complete node list can be found in S1 File.

and 1 = ON/high). The Boolean values were updated over discrete time steps according to logical rules derived from protein interactions as defined in the databases. The logical rules were initially assigned as follows.

1. An OR function was used to describe the updating rules of nodes with multiple activating regulators, e.g., $C(t+1) = A(t)$ OR $B(t)$, where $C(t+1)$ is the Boolean value of node C in the next time step, and $A(t)$ and $B(t)$ are the Boolean values of the activator nodes A and B in the current time step.

2. A NOT OR function was used to describe the updating rules of nodes with multiple inhibitory regulators, e.g., $C(t+1) =$ NOT $(A(t)$ OR $B(t))$. Here, node C turns on in the next time step only if nodes A and B are off in the current time step.

3. An inhibitory regulator wins over all activating regulators, e.g., $D(t+1) = (A(t)$ OR $B(t))$ AND NOT$(C(t))$, where nodes A and B are activating regulators while node C is an inhibitory regulator of node D.

These assumptions were initially made for the generic model in a manner similar to the previous studies [28–30]. However, more specific rules were derived when biological support was present. For instance, AKT is recruited by PIP3 at the plasma membrane. Subsequently, AKT is phosphorylated twice by MTORC2 and PDPK1 at the S473 and T308 sites, respectively, to be fully activated [39]. These interactions were described as

$$AKT\_i(\text{t} + 1) = PIP3(t) \text{ AND } MTORC2\_c(t) \text{ AND } PDPK1(t)$$

The complete Boolean rules of the generic model (Fig 2) can be found in S1 File.

## 3. Cell-line specific models

The generic Boolean model describes common signaling pathways related to cell growth and proliferation in breast cancer. However, the same combination of drugs can produce different synergistic effects on different cell lines. To account for these heterogenic responses, the generic Boolean model was tailored into cell-line specific models to capture each cancer cell line's unique characteristics and responses to drug perturbation. We created five cell-line specific models for MCF-7, T-47D, BT-549, MDA-MB-231, and MDA-MB-468 by

1. integrating cell line mutation profiles into the Boolean functions,

2. setting initial states of the cell lines based on gene expression profiles and

3. modifying the Boolean rules to match model simulations with gene expression data and observed behaviors from drug-perturbed experiments. These modifications are explained in detail below.

**3.1 Integrating cell line mutation profiles into the Boolean functions.**   The missense mutations for the five cell lines were identified from the Cell Model Passports (https://cellmodelpassports.sanger.ac.uk/) and DepMap (https://depmap.org/portal/) databases. Subsequently, the mutations were annotated for the biological function using the OncoKB database (https://www.oncokb.org/). We assigned a fixed Boolean value of 0 or 1 to nodes whose mutations were annotated as loss-of-function or gain-of-function, respectively. S1 Table displays the mutation profile of seven genes across the five breast cancer cell lines utilized in the present study.

**3.2 Setting initial states of the cell lines based on gene expression profiles.**   The gene expression data normalized with the transcripts per million (TPM) method was retrieved from Cell Model Passports. The normalized gene expression values were further scaled across five cancer cell lines by the Min-Max normalization method (Eq 1) to obtain values between 0 and 1:

$$x'_{jk} = \frac{x_{jk} - \min\left(x_j\right)}{\max\left(x_j\right) - \min\left(x_j\right)} \, , \tag{1}$$

where $x'_{jk}$ and $x_{jk}$ are the scaled and original expression values of gene $j$ from cell line $k$, respectively. $\max(x_j)$ and $\min(x_j)$ are the maximum and minimum, respectively, of the expression values of gene $j$ among five breast cancer cell lines. The scaled values (S1 Fig) were then used as a probability that the corresponding nodes were ON at the beginning of each simulation. For example, if the scaled value of gene $J$ of cell line $K$ is 0.7, the initial state of protein $J$ will be set to 1 (ON) with a 70% chance at the beginning of the simulation of cell line $K$.

**3.3 Modifying the Boolean rules to match model simulations with gene expression data and observed behaviors from drug-perturbed experiments.**   Next, each cell-line specific model (i.e., the generic model that incorporated genetic mutation and gene expression profiles) was simulated to generate protein activities at the steady state. The simulation results were compared to gene expression data and observed behaviors from drug-perturbed experiments of each respective cell line. During this step, we manually modified the Boolean rules of each cell-line specific model to capture observed behaviors derived from the experimental data.

To compare protein activities simulated from the Boolean model (values of 0 or 1) to gene expression data (values obtained from the Cell Model Passports database and scaled to range

between 0 and 1), we performed model simulation for 5000 time steps using a uniform asynchronous update scheme. In this scheme, only one node from all nodes that could change their state was selected with equal probability among others to change per time step. One hundred repeats were simulated for each cell-line specific model. The resulting Boolean values (0 or 1) of each protein from all repeats were averaged over the same time steps to obtain the protein activities (values ranging between 0 and 1), and the protein activities at the steady state were compared to the retrieved gene expression data from the Cell Model Passports database. The time step at which the steady state of the protein activities was reached was determined by entropy $H(t)$ [40]:

$$H(t) = -\sum_{S} P_t(S) \log_2(P_t(S)), \qquad (2)$$

where $P_t(S)$ is the probability the state $S$ is observed over time step 0 to $t$. The entropy was calculated across the simulation time step ($t = 0$ to 5000) to obtain the entropy profile. A higher entropy value indicates a higher impurity level in the system (i.e., various states of protein activities are observed over time $t$). When a particular state of protein activities becomes dominant over time, indicating the steady state, the entropy level approaches 0 as the probability of observing the steady state is near 1, i.e., $\lim_{P_t(S) \to 1} \log_2(P_t(S)) = 0$. However, it may take a long computational time to simulate the model until the system is completely homogenous with a steady state. Therefore, we assume that the protein activities have reached a steady state when the slope of the entropy profile is at its minimum. The protein activities from the time step the steady state was reached to the final simulation time step ($t = 5000$) were averaged to represent the steady-state protein activities.

Next, we collected 42 drug-treatment experiments on the five cell lines from published literature. We conducted a corresponding simulation for each experiment by fixing the Boolean values of the drug target nodes to 0 or 1, depending on whether the drug was an antagonist or agonist. We then compared the simulated response of the model to the observed behavior of the cell lines in the experiments.

During this step, inconsistency between simulations and experimental observations was addressed by manually modifying logical rules (e.g., changing the rule from AND to OR, and vice versa) and by adding/removing edges until the Boolean model of each cell line reproduced the gene expression and drug-treatment experiment observations. The adjustments were informed based on the literature as listed in S1 File.

For example, the generic Boolean function of MTORC1

$$MTORC1\_c(t+1) = RHEB(t) \qquad (3)$$

was modified into the ER-positive specific Boolean function

$$MTORC1\_c(t+1) = RHEB(t) \text{ OR } NRG1(t) \text{ OR } IGF1(t) \text{ OR } PIM(t) \qquad (4)$$

because, in the generic Boolean model, MTORC1 was activated only by RHEB, which is a downstream target of AKT (Eq 3). However, the regulatory function of MTORC1 of T-47D and MCF-7 further included NRG1, IGF1, and PIM (Eq 4). This modification was based on supporting evidence showing that the presence of NRG1, IGF1, or PIM could maintain the MTORC1 activity even when an inhibitor suppressed the AKT activity [9, 41].

In another instance, KMT2D, which is regulated positively and negatively by PAX7 and AKT, respectively, was described by a generic Boolean function

$$KMT2D\_c(t+1) = PAX7(t) \text{ AND NOT } AKT\_i(t) \tag{5}$$

The function was modified to

$$KMT2D\_c(t+1) = PAX7(t) \text{ OR NOT } AKT\_i(t) \tag{6}$$

in MCF-7 to reduce the contribution of AKT in inactivating KMT2D. The modification was made to match the simulation result with the observed expression data of KMT2D in MCF-7, which has a scaled value of 0.78 (out of the maximum value of 1.0). In contrast, for MDA-MB-231, the generic function in Eq 5 was used as the scaled expression data of KMT2D in this cell line was 0.

The complete logical rules for each cell-line specific model are listed in S1 File. The 42 experiments whose information was used to modify and validate the Boolean rules will be discussed further in the Results section (Table 2 below).

## 4. Drug combination simulations

We selected 150 drug combinations from the DrugComb database (https://drugcomb.fimm.fi/) to investigate using our cell-line specific models (S1 File). To simulate the drug combination effects, we identified the target proteins of 33 unique drugs in the dataset from Drugbank (https://go.drugbank.com/) and Therapeutic Target Database (TTD; https://db.idrblab.net/ttd/) (Table 1). Each cell-line specific model was simulated without drug perturbation until the steady state of protein activities was reached (Eq 2). Then, the effects of the drug combinations were applied. Four doses (0, 0.25, 0.75, and 1) of each drug in combination were investigated for each drug pair. The doses represented the probability of the target-protein nodes being OFF or ON (depending on whether the drug is an antagonist or agonist). For example, if fulvestrant (an ESR1 inhibitor) was used with a dose of 0.25, the ESR1 node had a probability of 0.25 to be OFF determined at the time the drug was applied to the model in each simulation repeat. Thirty simulation repeats were performed for each combination dose. The simulations continued until the steady state of protein activities was reached again.

Implementing drug effects on protein targets as a probability between 0 and 1 reflects the concentration ranges of monotherapy that cover the minimum and the maximum effect of drugs as implemented in the drug combination experiments [42–44]. We assume that a drug concentration with the maximum effect (equivalent to a dose of 1.0 in our simulation) would fully inhibit the protein targets with a 100% chance. Similarly, a drug concentration at half the maximum effect (equivalent to a dose of 0.5 in our simulation) would inhibit the protein targets with a 50% chance. However, as simulating various dose-response profiles can be computationally intensive, we have selected to use only four doses for each drug. As a result, the simulations produced 4×4 protein activity profiles per drug pair per cell line.

To calculate drug synergy scores from our model and compare them to the values reported in the DrugComb database, we proposed the calculation of the *PSS*. First, we categorized proteins in the model into onco- or tumor-suppressor proteins. Then, we calculated the *PSS* using the protein activities at the steady state, according to Eqs 7 and 8:

$$PSS_{\text{onco-protein}} = \min\left( X_1^{\text{SS}}, X_2^{\text{SS}} \right) - X_{1+2}^{\text{SS}} \tag{7}$$

$$PSS_{\text{tumor-suppressor protein}} = Y_{1+2}^{\text{SS}} - \max\left( Y_1^{\text{SS}}, Y_2^{\text{SS}} \right), \tag{8}$$

**Table 1. Drug target proteins identified from Drugbank and Therapeutic target database.**

| Drug | Drug target proteins (Asterisks indicate an agonistic effect; Non-asterisks indicate an antagonistic effect.) |
|---|---|
| 391210-10-9, SELUMETINIB, TRAMETINIB | MEK1, MEK2 |
| 915019-65-7 (Dactolisib) | PI3KCA, MTORC1, MTORC2 |
| ALPELISIB, BUPARLISIB | PI3KCA |
| ANTIBIOTIC AY 22989, DEFOROLIMUS, NSC733504, TEMSIROLIMUS | MTORC1 |
| AZD2014 | MTORC1, MTORC2 |
| AZD5363 | AKT1, AKT3 |
| CELECOXIB | PDPK1 |
| DASATINIB | SRC, LCK, ABL1, ABL2 |
| EMCYT, MITOTANE, RALOXIFENE | ESR1* |
| ERLOTINIB HYDROCHLORIDE, GEFITINIB, VANDETANIB | EGFR |
| FULVESTRANT | ESR1 |
| IMATINIB, NILOTINIB | ABL1 |
| LAPATINIB | EGFR, HER2 |
| MEGESTROL ACETATE | PGR* |
| MK-2206 | AKT3 |
| PACLITAXEL | BCL2 |
| RUXOLITINIB | JAK2 |
| SAPITINIB | HER2 |
| SORAFENIB | EGFR, RAF1 |
| STATTIC | STAT3 |
| TAMOXIFEN CITRATE | ESR1, MAPK8* |
| TRISENOX | JUN*, CCND1, MAPK3*, MAPK1*, AKT1* |

where $X^{SS}$ and $Y^{SS}$ represent the steady-state protein activity of an oncoprotein and tumor-suppressor protein, respectively. The subscripts 1, 2, and 1+2 indicate the steady-state protein activity after being perturbed by drug 1, drug 2, and both, respectively. The calculation of *PSS*s for each dose followed the methodology for computing the highest single agent (HSA) synergy score. The value of the *PSS* is between −1 and +1, where the positive and negative values were interpreted as the drug pair having a synergistic and antagonistic effect on the protein level, respectively. The *PSS* of 0 indicated the additive effect. The *PSS* at the 75th percentile across all dose combinations was designated the consensus *PSS* for that protein. Finally, the *PSS*s of selected proteins were summed together to represent the predicted HSA synergy score for each drug pair.

To determine which proteins whose *PSS*s should be used to calculate the predicted HSA score, a genetic algorithm was implemented to find the best representatives from both onco- and tumor-suppressor proteins by the following steps:

1. The initial population in the genetic algorithm comprised 60 chromosomes with a random length (the number of representative proteins) and randomly chosen proteins, except E2F1 and CASP3, which were selected by default.

2. The predicted HSA scores of the 150 drug pairs were calculated from each chromosome by summing the *PSS*s of all proteins in the chromosome. Then, the chromosomes were evaluated by their resulting Pearson's correlation coefficient between the predicted and observed

HSA synergy scores. The top 30 chromosomes that yielded the highest Pearson's correlation coefficients were selected.

3. Genetic mutation and crossing-over were applied both with the probability of 0.9 to the selected chromosomes to create 30 offspring chromosomes. The high probability values (0.9) were assigned due to the large search space.

4. New 30 chromosomes were created and combined with the offspring chromosomes to constitute the new population.

5. The processes in steps 2–4 were repeated for 10,000 generations to ensure the optimally selected protein representatives.

## 5. Mechanism explanations

Finally, the potential synergistic and antagonistic mechanisms of drug combinations were investigated by clustering drug pairs into groups based on the similarity in the selected *PSS*s. Through an analysis of the *PSS* profile patterns, we identified the contributions of individual protein activities to the observed synergy of drug pairs, characterizing them as distinct modes of drug synergism.

## Results

### 1. Protein signaling network reconstruction

Using the STRINGdb API (https://pypi.org/project/stringdb/), the majority of the original 76 signaling proteins from the KEGG database were significantly enriched in the PI3K-AKT (hsa04151: $N = 51$ FDR = $8.74 \times 10^{-64}$) and MTOR (hsa04150: $N = 29$, FDR = $7.05 \times 10^{-50}$) signaling pathways.

When 41 more nodes from the SIGNOR and SignaLink databases were included in the network, more pathways showed significant enrichment, e.g., apoptosis (hsa04210: $N = 21$ from KEGG and 8 from SIGNOR and SignaLink, FDR = $3.46 \times 10^{-42}$), ERBB (hsa04012: $N = 22$ from KEGG and 8 from SIGNOR and SignaLink, FDR = $4.04 \times 10^{-48}$), MAPK (hsa04010: $N = 22$ from KEGG and 9 from SIGNOR and SignaLink, FDR = $3.34 \times 10^{-33}$), and FOXO (hsa04068: $N = 29$ from KEGG and 2 from SIGNOR and SignaLink, FDR = $7.49 \times 10^{-46}$) signaling pathways.

Overall, the reconstructed protein signaling network comprehensively covered multiple breast cancer growth and proliferation pathways, including the PI3K-AKT, MTOR, apoptosis, ERBB, MAPK, and FOXO signaling pathways.

### 2. Model validation

The five cell-line specific models were simulated with and without drug perturbations to obtain protein activities at the steady state. The unperturbed protein activities from each cell line were compared to their respective cell lines' actual gene expression data retrieved from the Cell Model Passports database. The drug perturbation simulations were compared to observed responses of the cell lines to drug treatments collected from 42 experiments. The results of these simulations demonstrated that the models could capture the unique characteristics of each cell line and their responses to drug perturbations.

**2.1 Model simulation compared to gene expression profiles.** We conducted simulations for each cell-line specific Boolean model without drug perturbations to derive the steady-state protein activities. The simulated steady-state protein activities exhibited a reasonable level of

correlation with the actual gene expression data, with Pearson's correlation coefficients ranging between 0.67 and 0.85. MDA-MB-231 and T-47D were the cell lines from triple-negative and ER-positive subtypes with the highest correlation at 0.85 and 0.75, respectively. An identical Pearson's correlation at 0.67 was observed for MDA-MB-468 and BT-549, whereas MCF-7 was achieved at 0.68. When we performed clustering analysis, combining these simulated steady-state protein activities with the actual gene expression data, the results accurately segregated into the ER-positive (T-47D and MCF-7) and triple-negative (BT-549, MDA-MB-231, and MDA-MB-468) sub-groups, as demonstrated in Fig 3. This suggests that our cell-line specific models behaved similarly to their unperturbed cell-line behaviors.

**2.2 Model simulation compared to drug-treatment experiments.** The Boolean model of each cell line was further validated by drug-treatment experiments from the literature. Table 2 shows that the models captured many drug-resistance behaviors, mainly involving the release of negative regulation of an oncogenic pathway. We used the activities of E2F1 and CASP3 in representing cell proliferation and apoptosis in response to single-drug and combination treatments.

Some examples of simulations from Table 2 are illustrated in Fig 4. Fig 4A demonstrates the release of negative regulation on the MAPK (Ras-Raf-MEK-ERK) pathway from MTORC1 inhibition (rapamycin) in BT-549, leading to rapamycin resistance [8]. In Fig 4B, MTORC1 inhibition released the inhibition on IRS1, triggering the activation of AKT in MCF-7 [45]. In Fig 4C, a MEK inhibitor removed the inhibition on EGFR by ERK, leading to the activation of the PI3K/AKT pathway in MDA-MB-231 [47]. Fig 4D plots the simulation of a resistance mechanism in MDA-MB-468 in response to a PI3K inhibitor. The inhibition of PI3K released the negative regulation of the IRS1 protein by the AKT-MTORC1-S6K1 pathway, which subsequently activated the JAK2-STAT5 cascade, allowing the cells to escape cell death [10].

## 3. Drug synergy predictions

Next, we used the validated cell-line specific models to calculate the *PSS*s for 150 drug pairs retrieved from the DrugComb database. The *PSS*s of 13 nodes (BCL2, CYCLIN D, MTORC1, ESR1, PGR, PAK1, STAT3, WNT1, BID, GATA3, FADD, P21, and CASP3) were selected by the genetic algorithm to calculate the predicted HSA synergy score (see Methods). The selected 13 nodes contribute to different breast cancer-related pathways: cell cycle (CYCLIN D and P21), apoptosis (BCL2, BID, CASP3, and FADD), estrogen signaling (ESR1), EGFR signaling (PAK1), progesterone signaling (PGR), Wnt signaling (WNT1), JAK/STAT signaling (GATA3 and STAT3), and PI3K/AKT signaling (MTORC1) pathways. The comparison between the predicted HSA synergy scores and the experimental HSA scores from MDA-MB-231 achieved the highest Pearson's correlation coefficient of 0.58 with a great balance among the classification metrics (AUC = 0.74, sensitivity = 0.63, and specificity = 0.64). When considering all five cell-line specific models, the models achieved a Pearson correlation coefficient of 0.326 (Fig 5), an AUC of 0.62 with a sensitivity of 0.50, and a specificity of 0.67 (Fig 6).

## 4. Predicted mechanisms

Finally, the potential synergistic and antagonistic mechanisms of drug combinations were analyzed by clustering the drug pairs into groups based on the 13 selected *PSS*s. For the synergistic group, 42 drug pairs, correctly predicted as true positives, from five cell lines were categorized into five groups (Fig 7). For groups 1 and 2, the synergistic effect mainly relied on the induction of the apoptosis pathway (indicated by a positive *PSS* of CASP3). In group 1, the extrinsic apoptosis pathway was synergistically upregulated (positive *PSS*s of FADD and BID), while the intrinsic pathway was synergistically induced in group 2 (a positive *PSS* of BCL2). An example

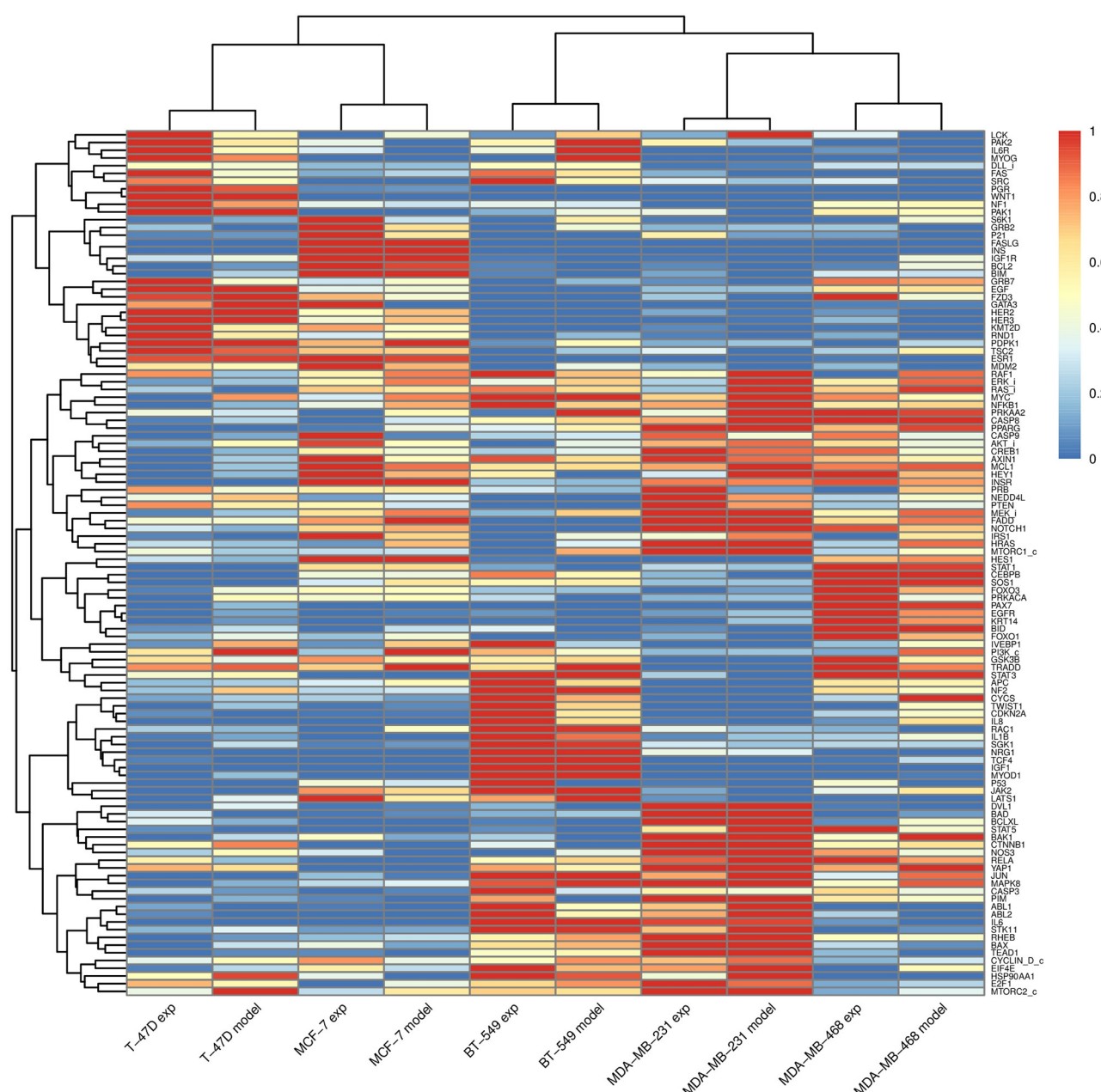

**Fig 3. The simulated steady-state protein activities of each cell line.** The steady-state protein activities simulated from the five cell-line specific Boolean models were clustered with the actual gene expression profiles (labeled as 'exp' in the figure) based on the correlation distance. The actual gene expression profiles were scaled with the Min-Max normalization method to yield a value between 0 and 1 before clustering.

of a drug pair from group 1 was AZD5363 + MK-2206. MK-2206 (AKT3 inhibitor) alone can induce apoptosis in T-47D [52].Combining MK-2206 with AZD5363 (AKT1 and AKT3 inhibitor) resulted in synergistic activation of apoptosis mainly from the extrinsic pathway. However, the combination also activated ESR1 and CYCLIN D (the negative values of *PSSs* in Fig 7) due to the release of negative regulation on FOXO3 by AKT, which is consistent with experimental observations [7].

**Table 2. Drug-treatment simulations from cell-line specific models.**

| Treatment | Cell line | E2F1 | CASP3 | Simulation description in alignment with the experiments | Reference |
|---|---|---|---|---|---|
| - | MCF-7 | 0.36 | 0.02 | Simulation of unperturbed MCF-7 | - |
| - | T-47D | 0.56 | 0.07 | Simulation of unperturbed T-47D | |
| - | MDA-MB-231 | 0.85 | 0.42 | Simulation of unperturbed MDA-MB-231 | |
| - | MDA-MB-468 | 0.27 | 0.42 | Simulation of unperturbed MDA-MB-468 | |
| - | BT-549 | 0.72 | 0.52 | Simulation of unperturbed BT-549 | |
| Rapamycin | MCF-7 | 0 | 0 | MTORC1 inhibition by rapamycin releases the inhibition on IRS1, which activates the Ras-Raf1-MEK1/2-ERK pathway in MCF-7 and BT-549 (Fig 4A). Combining UO126 (a MEK1/2 inhibitor) with rapamycin disrupts the resistance mechanism. The combination both decreases proliferation and increases apoptosis. | [8] |
| Rapamycin + UO126 | | 0.02 | 0.3 | | |
| Rapamycin | BT-549 | 0 | 0.11 | | |
| Rapamycin + UO126 | | 0.01 | 1.0 | | |
| Rapamycin | MCF-7 | 0 | 0 | MTORC1 inhibition by rapamycin releases the inhibition on IRS1, triggering the activation of AKT in MCF-7 (Fig 4B) and MDA-MB-468. Combining rapamycin and NVP-AEW541 (an inhibitor of IGF1R, an upstream activator of IRS1) reduces the AKT level in both cell lines. However, the suppression of AKT activates ESR1 via FOXO3, which induces the proliferation of MCF-7. | [45] |
| Rapamycin + NVP-AEW541 | | 1 | 0.15 | | |
| Rapamycin | MDA-MB-468 | 0 | 0.05 | | |
| Rapamycin + NVP-AEW541 | | 0 | 0.15 | | |
| BYL719 | MCF-7 | 0.95 | 0.32 | The inhibition of PI3K leads to the upregulation of the FOXO3 transcription factor and ESR1 in MCF-7 and T-47D. The combination of BYL719 (a PI3K inhibitor) and fulvestrant (an ESR1 inhibitor) can mitigate the resistance. | [7] |
| BYL719 + FULVESTRANT | | 0 | 0.8 | | |
| BYL719 | T-47D | 0.86 | 0.5 | | |
| BYL719 + FULVESTRANT | | 0.41 | 0.40 | | |
| BYL719 | MCF-7 | 0.99 | 0.34 | BYL719 (a PI3K inhibitor) alleviates AKT-induced repression on KMT2D, leading to the upregulation of ESR1 in the ER+ cell lines. Combining a KMT2D inhibitor with the PI3K inhibitor results in a synergistic outcome. | [46] |
| BYL719 + KMT2D inhibitor | | 0 | 0.34 | | |
| BYL719 | T-47D | 0.92 | 0.54 | | |
| BYL719 + KMT2D inhibitor | | 0.33 | 0.23 | | |
| BYL719 + NRG1 | MCF-7 | 1 | 0 | BYL719 suppresses the activities of PI3K and AKT, but the presence of NRG1 maintains the MTORC1 level in MCF-7 and T-47D. Combining BYL719 with RAD001 (an MTOR inhibitor) results in a synergistic effect. | [41] |
| BYL719 + RAD001 + NRG1 | | 0 | 0 | | |
| BYL719 + NRG1 | T-47D | 1 | 0.44 | | |
| BYL719 + RAD001 + NRG1 | | 0.86 | 0.55 | | |
| BYL719 + IGF1 | MCF-7 | 1 | 0 | BYL719 suppresses the activities of PI3K and AKT, but the presence of IGF1 maintains the MTORC1 level in MCF-7 and T-47D. Combining BYL719 with RAD001 (an MTOR inhibitor) results in a synergistic effect. | |
| BYL719 + RAD001 + IGF1 | | 0 | 0 | | |
| BYL719 + IGF1 | T-47D | 1 | 0 | | |
| BYL719 + RAD001 + IGF1 | | 0.5 | 0 | | |
| BYL719 | T-47D | 0.92 | 0.54 | BYL719 suppresses the activities of PI3K and AKT. However, PIM sustains the MTORC1 level due to the shared target between PIM and AKT. Therefore, the combination of PI3K and PIM inhibitors mitigates the resistance. | [9] |
| BYL719 + PIM inhibitor | | 0.79 | 0.41 | | |
| U0126 | T-47D | 0.55 | 0.09 | U0126 suppresses MEK and ERK, triggering the inhibition release on EGFR, which leads to the upregulation of PI3K in MDA-MB-231 (Fig 4C). Therefore, combining a PI3K inhibitor (e.g., PIK90) with U0126 results in a synergistic outcome. | [47] |
| U0126 + PI3K inhibitor | | 0.93 | 0.45 | | |
| U0126 | MDA-MB-231 | 0 | 0.43 | | |
| U0126 + PI3K inhibitor | | 0 | 0.47 | | |
| BEZ235 | MDA-MB-468 | 0 | 0.18 | PI3K inhibition by BEZ235 increases the IRS1-dependent activation of the JAK2/STAT5 signaling pathway and the secretion of IL-8 in MDA-MB-468 (Fig 4D). The combination between BEZ235 and a JAK2 inhibitor (e.g., NVP-BSK805) disrupts the resistance. | [10] |
| BEZ235 + JAK2 inhibitor | | 0 | 1 | | |

*(Continued)*

**Table 2.** (Continued)

| Treatment | Cell line | E2F1 | CASP3 | Simulation description in alignment with the experiments | Reference |
|---|---|---|---|---|---|
| Dasatinib | MDA-MB-468 | 0.23 | 0.47 | By treatment with dasatinib, the AKT level is still maintained in MDA-MB-468. The combination of dasatinib and MK-2206 (an AKT inhibitor) results in a synergistic outcome. | [48] |
| Dasatinib + MK-2206 | | 0.01 | 0.07 | | |
| Gefitinib | MDA-MB-468 | 0.31 | 0.79 | The combination between MK-2206 (an AKT inhibitor) and gefitinib (an EGFR inhibitor) synergistically reduces the proliferation of EGFR-inhibitor resistant MDA-MB-468. | [49] |
| Gefitinib + MK-2206 | | 0.05 | 0.56 | | |
| Gefitinib | MDA-MB-231 | 0.83 | 0.67 | The combination of RPTOR siRNA and gefitinib (an EGFR inhibitor) synergistically reduces the proliferation of EGFR-inhibitor resistant MDA-MB-231 | [49] |
| Gefitinib + RPTOR siRNA | | 0 | 0.43 | | |
| Gefitinib + AT7867 | MDA-MB-231 | 0 | 0.43 | The activity of the RAS-RAF1-MEK-ERK pathway is sustained in MDA-MB-231 treated with gefitinib (an EGFR inhibitor) + AT7867 (an AKT inhibitor). Combining PD-0326901 (a MEK inhibitor) as a triple treatment synergistically induces apoptosis. | [50] |
| Gefitinib + AT7867 + PD-0326901 | | 0 | 0.6 | | |
| Paclitaxel | MDA-MB-231 | 0.8 | 0.63 | The combination of paclitaxel (a BCL2 inhibitor) and everolimus (an MTOR inhibitor) achieves a synergistic outcome in MDA-MB-231. | [51] |
| Paclitaxel + Everolimus | | 0.02 | 0.53 | | |

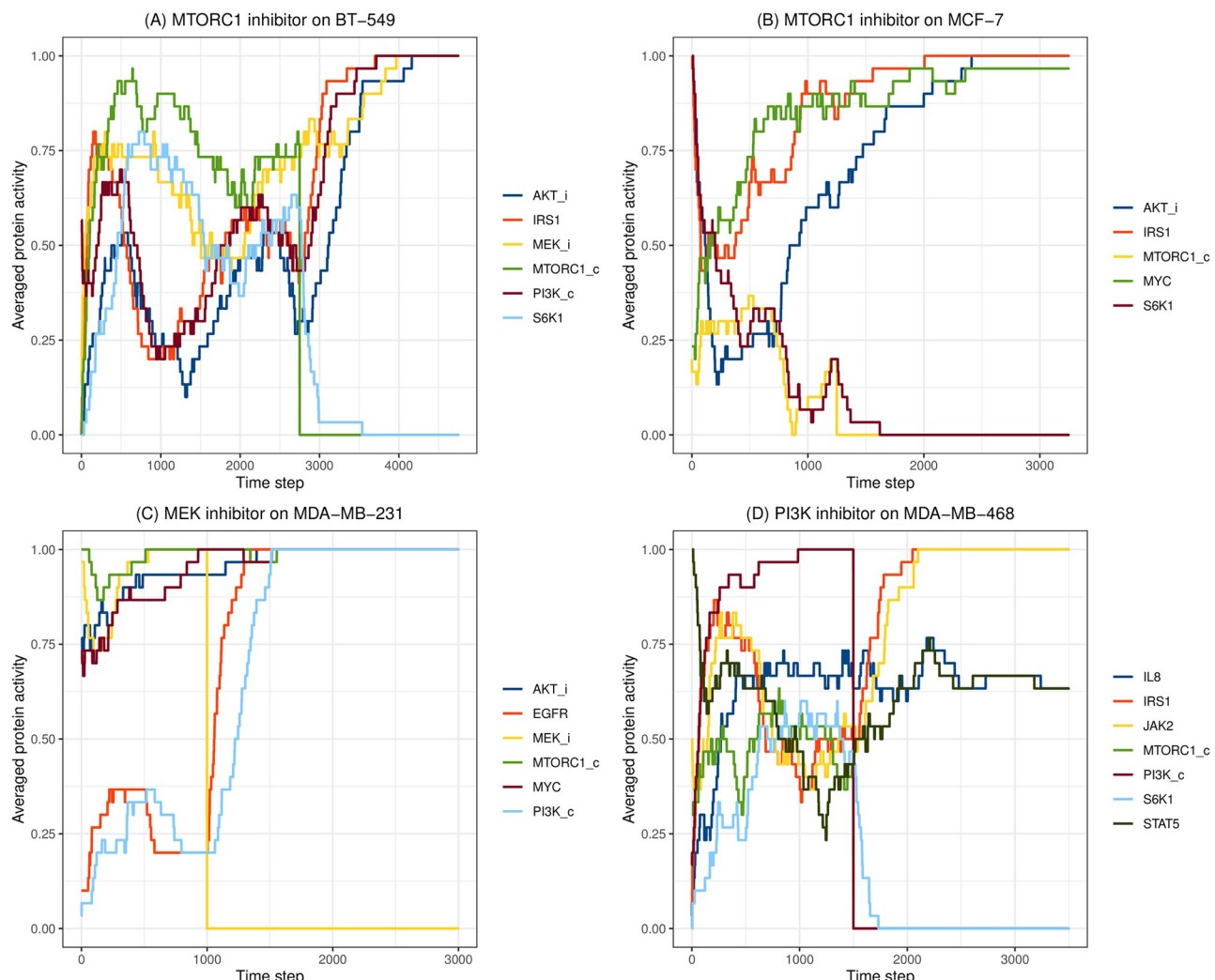

**Fig 4. Simulations of resistance mechanisms in response to drug treatments.** (A) MTORC1 inhibition in BT-549. (B) MTORC1 inhibition in MCF-7. (C) MEK inhibition in MDA-MB-231. (D) PI3K inhibition in MDA-MB-468.

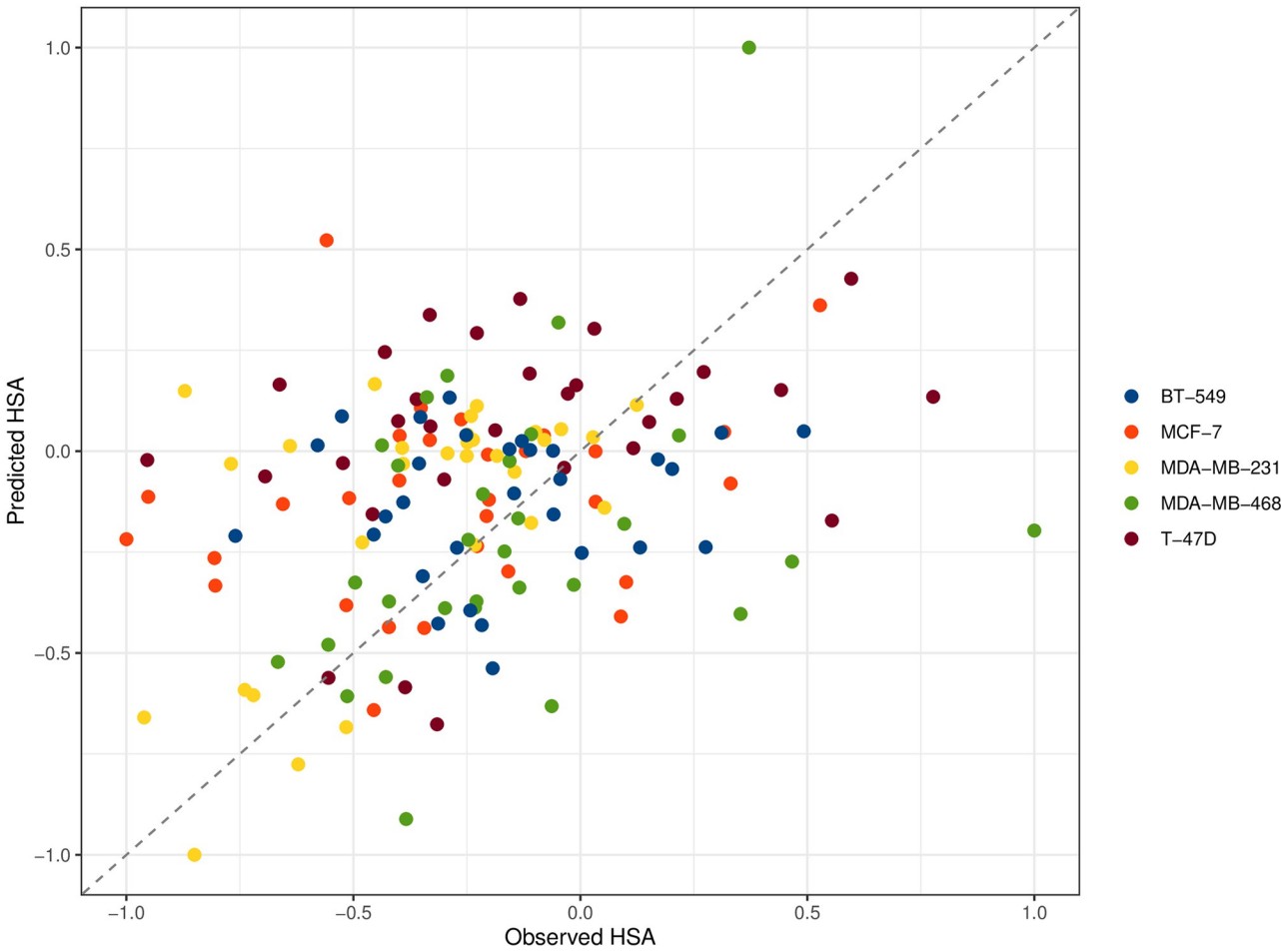

**Fig 5. Scatter plot between actual and predicted HSA synergy scores.** The actual and predicted HSA scores were scaled with the Min-Max normalization method to achieve a score between −1 and +1.

An example of a drug combination in group 2 included MK-2206 (AKT inhibitor) + selu-metinib (MEK inhibitor), strongly inducing the intrinsic apoptosis pathway in MCF-7. It is possible that selumetinib increased the sensitivity of cells to MK-2206 in inducing intrinsic apoptosis because MK-2206 alone could not effectively cause apoptosis in MCF-7 when com-pared to unperturbed cells [53]. However, the negative effect on CYCLIN D was also observed due to the release of the inhibition of FOXO3 by AKT [7]. In MDA-MB-468, the combination of erlotinib hydrochloride (EGFR inhibitor) and imatinib (ABL1 inhibitor) in group 2 increased the intrinsic apoptosis and reduced the $G_1$/S transition and protein translation (indi-cated by the positive *PSS*s of CYCLIN D and MTORC1, respectively). This predicted mecha-nism aligns with a previous report involving the combination of a different EGFR inhibitor (lapatinib) and imatinib (an ABL1 inhibitor), where it was observed that the level of MYC, a downstream target of MTORC1, exhibited a synergistic reduction in MDA-MB-468 cells [54].

For groups 3 to 5, the proliferation was mainly inhibited (indicated by the positive *PSS*s of CYCLIN D and/or MTORC1). An example of a drug pair in group 3 was MK-2206 + AZD5363 (both target the AKT proteins) treated in BT-549. Unlike the effect of the same drug combination in the ER-positive T-47D cell line, the drug pair synergistically reduced the CYCLIN D activity in BT-549. Drug pairs in group 4 were the combination of an estrogen

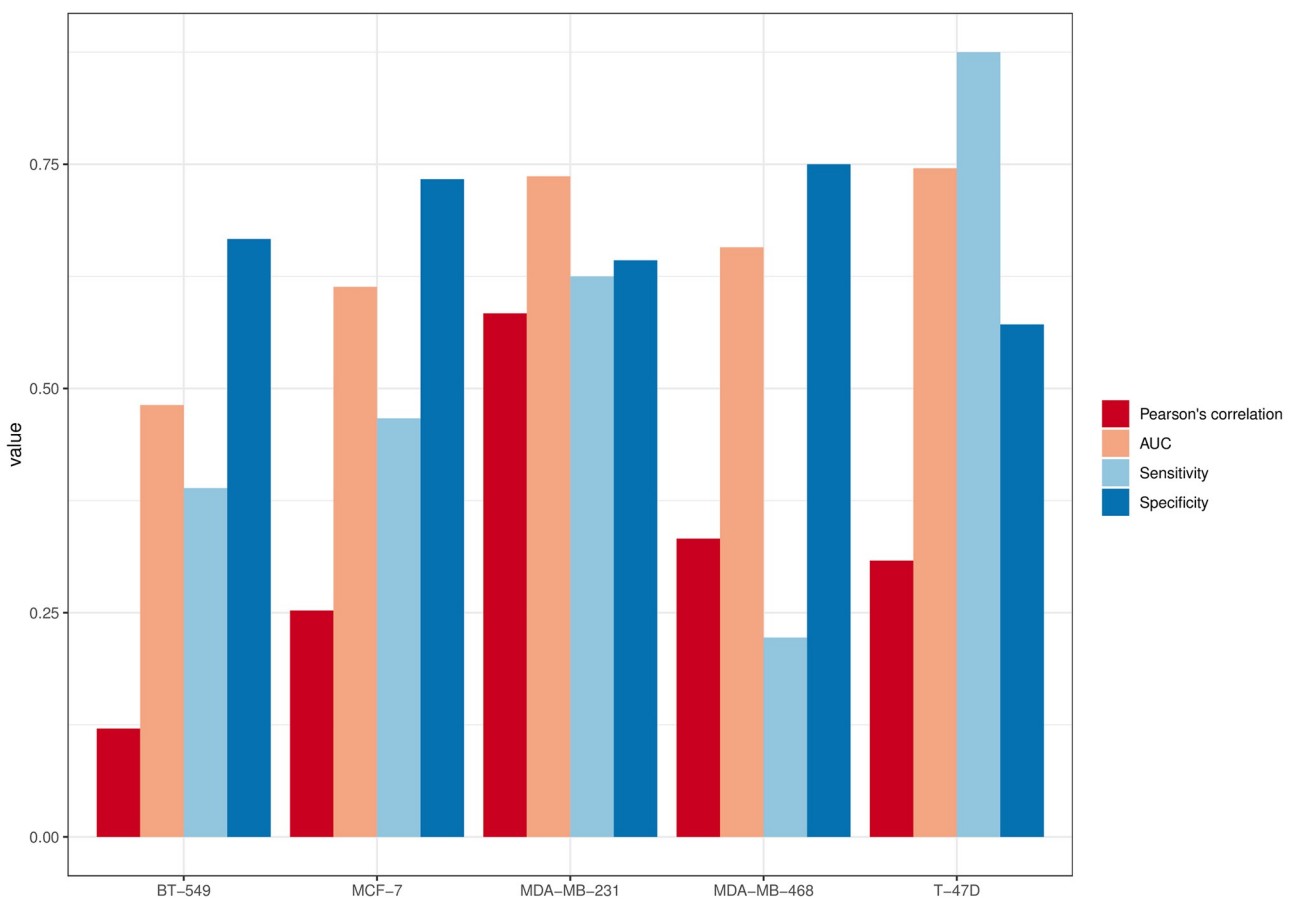

**Fig 6. Prediction performance of the models.** Pearson's correlation coefficients, AUC, sensitivity, and specificity scores of the cell-line specific models are shown.

receptor inhibitor (tamoxifen citrate or fulvestrant) and another drug. The combinations reduced the $G_1$/S transition by synergistically inhibiting ESR1 and activating P21. The synergy of drug pairs in group 5 was contributed mainly by the down-regulation of CYCLIN D. An example was an EGFR inhibitor (gefitinib or lapatinib) in combination with antibiotic AY 22989 (MTORC1 inhibitor). The combination of EGFR and MTOR inhibitors was previously reported with a synergistic effect in MDA-MB-231 [49]. We predict the mechanism to be the synergistic reduction in $G_1$/S transition and protein translation (due to the observed positive *PSS*s of CYCLIN D and MTORC1).

Table 3 lists seven drug pairs whose synergistic predictions were supported by in vitro assays from the literature. Here, we predict the synergistic contributions made by individual proteins and list them in Table 3.

For the antagonistic group, Fig 8 shows three separate groups of 45 drug pairs, correctly predicted as true negatives. In group 1, many drug pairs contained megestrol acetate, which has an agonist effect on the hormone receptor PGR. Thus, drugs combined with megestrol acetate had a highly negative value of *PSS* in PGR (e.g., vandatinib + megestrol acetate in MCF-7 and ruxolitinib + megestrol acetate in MDA-MB-231).

In group 2, most drug pairs contained an ESR1 agonist (emcyt, mitotane, or raloxifene). Therefore, drug pairs in this group exhibited a highly negative *PSS* value of ESR1 (e.g., gefitinib

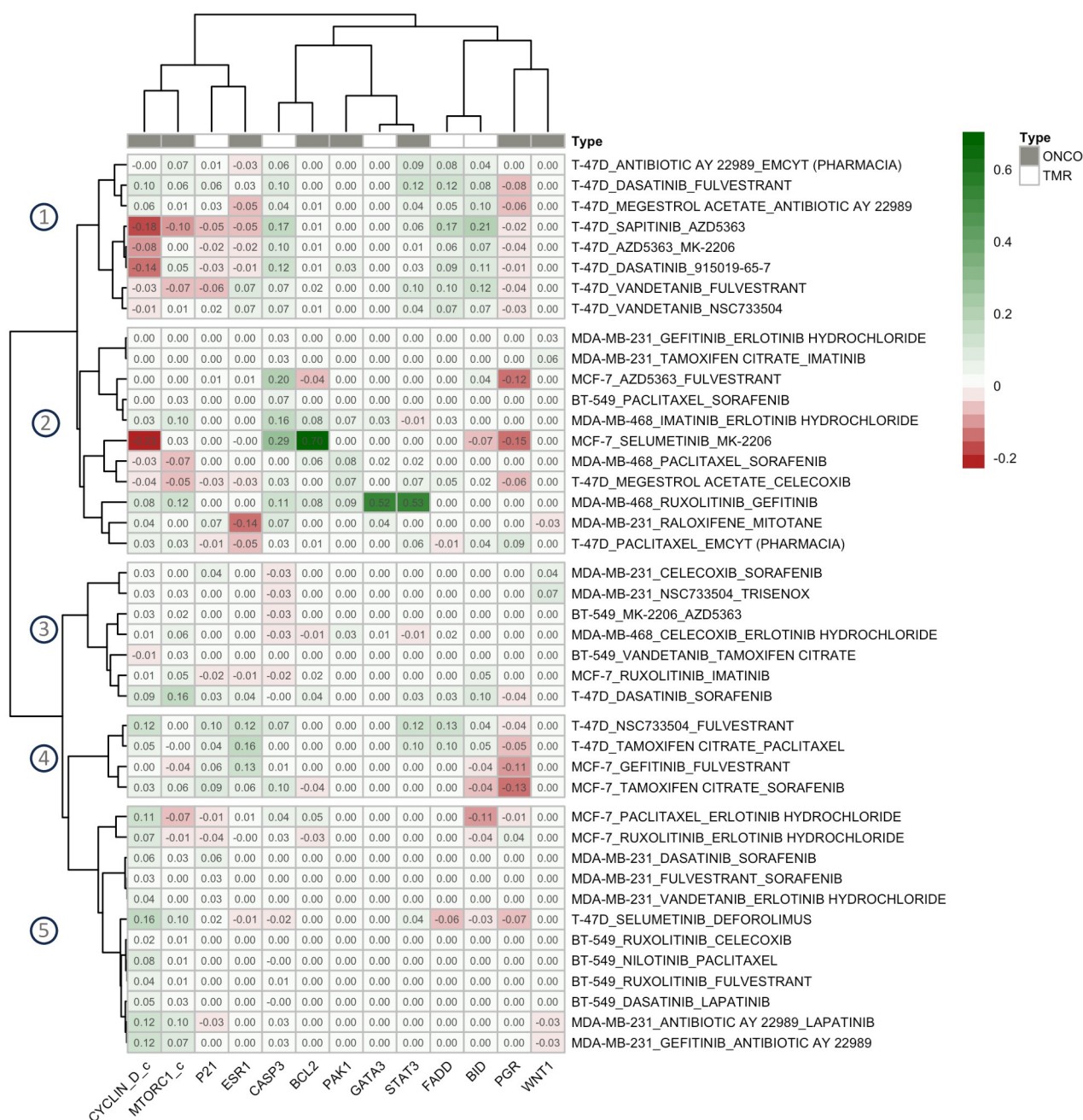

**Fig 7. Clustering analysis of true-positive synergistic drug pairs based on the protein synergy scores of the genetic algorithm-selected proteins.** The positive scores (green) indicate increased or decreased protein activities due to the drug combination perturbation compared to the single-drug treatment for tumor-suppressor or oncoproteins, respectively. The negative scores (red) indicate vice versa.

+ emcyt in MDA-MB-468 and raloxifene + imatinib in MDA-MB-231 or nilotinib in BT-549). Finally, we observed in group 3 that most drug pairs failed to inhibit proliferation nor promote apoptosis (indicated by the negative *PSS*s of MTORC1 and CASP3). For example, trisenox, which has an agonistic effect on JUN, MAPK3, MAPK1, and AKT1, combined with either erlotinib hydrochloride or megestrol acetate in MDA-MB-468 exhibited the highly negative *PSS*s of MTORC1 and CASP3.

**Table 3. Synergistic drug pairs with the literature support.**

| Drug Pair | HSA | | Proteins with predicted positive *PSSs* | |
|---|---|---|---|---|
| | **Pred** | **Exp** | **Apoptosis pathway** | **Proliferation pathway** |
| NSC733504 + Fulvestrant on T-47D [55] | 6.65 | 16.7 | CASP3 FADD BID | CYCLIN D STAT3 ESR1 P21 |
| Dasatinib + Fulvestrant on T-47D [56] | 5.90 | 2.57 | CASP3 FADD BID | CYCLIN D STAT3 P21 MTORC1 |
| AZD5363 + Fulvestrant on MCF-7 [57] | 1.05 | 11.34 | CASP3 BID | - |
| Gefitinib + Fulvestrant on MCF-7 [58] | 0.26 | 2.82 | - | ESR1 P21 |
| Dasatinib + Sorafenib on MDA-MB-231 [59] | 1.50 | 0.46 | - | CYCLIN D MTORC1 P21 |
| AY22989 + Lapatinib on MDA-MB-231 [60] | 2.21 | 7.58 | - | CYCLIN D MTORC1 |
| AY22989 + Gefitinib on MDA-MB-231 [49] | 2.17 | 0.71 | - | CYCLIN D MTORC1 |

## Discussion and conclusions

Finding drug pairs for effective breast cancer treatment is challenging due to a large combination space that requires resources and time to investigate. In the present study, we utilized a Boolean modeling approach to capture the drug-response behaviors of cancer cells and predict the synergistic effects between drug pairs. Our approach shares similarities with the methodology employed by Tsirvouli et al. (2020) [28] but extends the analysis to five breast cancer cell lines covering both ER-positive and triple-negative subtypes of breast cancers. In addition, a unique feature in the current study is the calculation of the protein synergy scores (*PSSs*) that determine the effect of drug combinations on individual proteins' activities. The calculation allowed the synergistic effects of drug pairs to be decomposed into the contributions made by individual proteins, thereby enabling the interpretation of the mechanism of the synergy through the profile of protein activities.

Our results identified five patterns of synergy among drug pairs: 1) synergistic activation of FADD and BID (extrinsic apoptosis pathway), 2) synergistic inhibition of BCL2 (intrinsic apoptosis pathway), 3) synergistic inhibition of MTORC1, 4) synergistic inhibition of ESR1, and 5) synergistic inhibition of CYCLIN D. Interestingly, the combination of AZD5363 and MK-2206 exhibited distinct synergistic mechanisms in T-47D and BT-549. In T-47D, it synergistically activated the extrinsic apoptosis pathway, while in BT-549, it prompted a synergistic inhibition of the proliferation pathway. Furthermore, our models have indicated that antagonistic interactions among drug pairs predominantly arise from the agonistic properties of specific drugs, which activate proteins such as ESR1, PGR, AKT, and ERK. Consequently, it is important to account for these factors during the development and evaluation of drug combinations to ensure their efficacy.

Our current study focused on simulating combinations of only two drugs because most drug-perturbed experiments involved only drug pairs. In addition, the HSA synergy scores from the DrugComb database to which we compared our synergy predictions were also limited to two drugs. However, cancer patients can be treated with more than two drugs if they

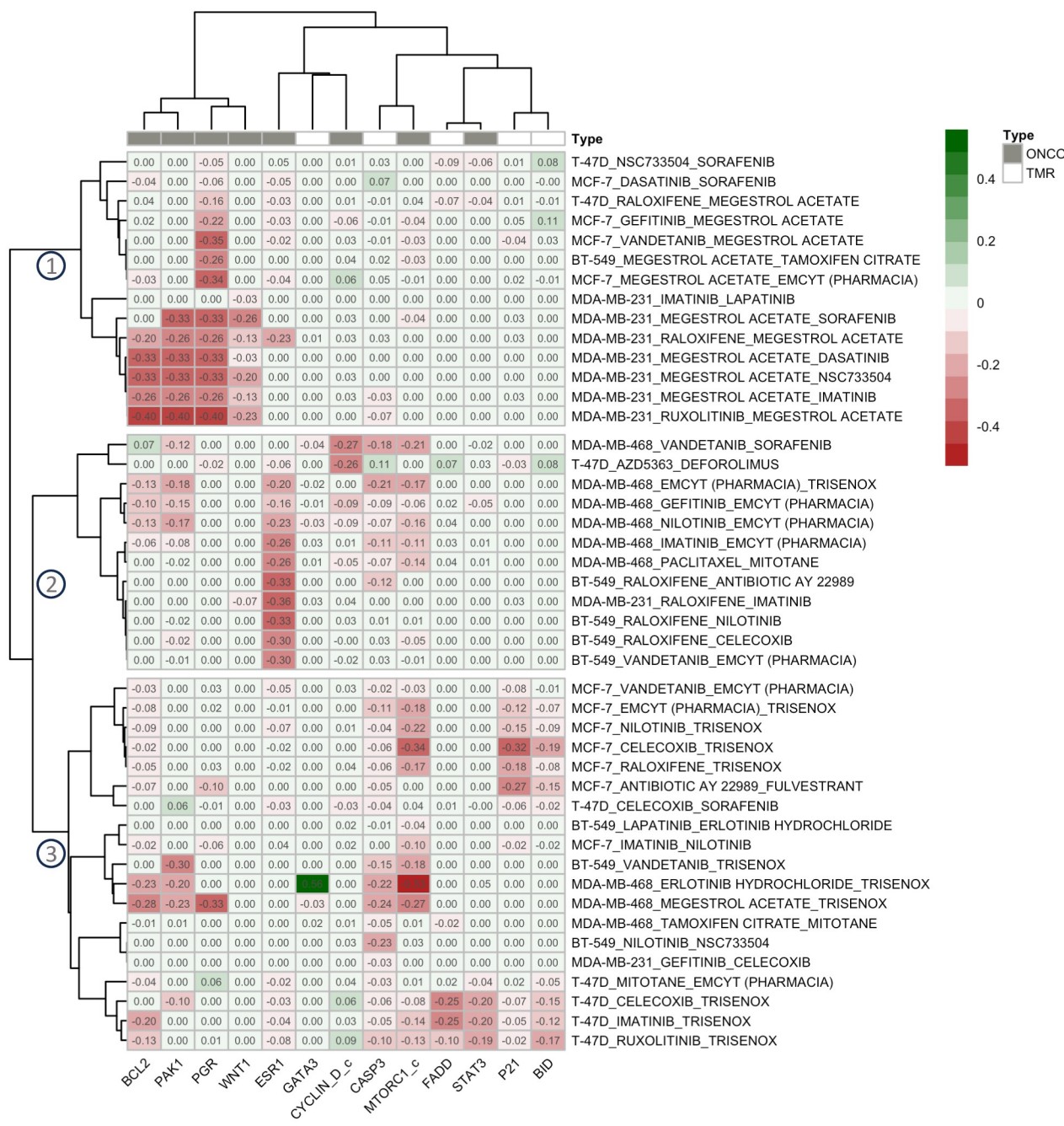

**Fig 8. Clustering analysis of true-negative antagonistic drug pairs based on the protein synergy scores of the genetic algorithm-selected proteins.** The positive scores (green) indicate increased or decreased protein activities due to the drug combination perturbation compared to the single-drug treatment for tumor-suppressor or oncoproteins, respectively. The negative scores (red) indicate vice versa.

offer greater efficacy [12], help to overcome drug resistance [2], and avoid drug toxicity [61]. It is worth noting that our simulation framework allows the simulation of more than two drug combinations, thereby providing further valuable predictions, which are planned for future investigation of the study.

We acknowledge that there is room for significant improvements in the future development of our models. First, the models require the integration of additional drug resistance mechanisms associated with ESR1 and EGFR. These proteins were frequently present as drug-pair targets, resulting in false-positive predictions (S2 Fig). By incorporating the resistance mechanisms, cancer cells may enhance their ability to counteract the effects of drugs, shifting the false positive to true negative predictions. For the false negative predictions, we speculate that including more experimental observations for each specific drug could help rectify this issue. At present, the validation of our models relied on the effects of only a limited number of drugs per cell line (as indicated in Table 2), which differs from all simulated drugs used in synergy predictions (as listed in Table 1).

Second, we utilized the highest single agent (HSA) model among the available synergy scores to quantify the drug pair's synergistic effect. HSA has been widely used for drug synergy prediction in many types of cancer due to the simplest assumption [28–30]. It should be noted, however, that the synergy model selection depends on individual preference [62], and the conflict among them can be observed (see examples in S2 Table). To reduce the misclassification from a single synergy score metric, a method capable of prioritizing synergistic drug pairs based on the consensus of all synergy score metrics is needed.

Third, integrating phosphoproteomic data would significantly enhance the model's performance. The genomic and transcriptomic data have been used for transforming a generic Boolean model into a cell-line specific one due to its abundant availability [63, 64]. However, protein signaling networks are predominately influenced by the enzymatic activity of signaling proteins. Thus, the phosphoproteomic data should be used more precisely in the model development.

Additionally, the use of asynchronous updates in the current study did not consider the varying time scales of different reactions (e.g., protein synthesis versus phosphorylation/dephosphorylation). Incorporating this aspect into future versions of the model may enhance the simulation's accuracy. Furthermore, it should be mentioned that the five cell lines currently selected for the study are not representative of all subtypes of breast cancer cells. Among the five cell lines, three are triple-negative breast cancer (TNBC), and two are estrogen receptor-positive (ER-positive). This limited selection of cell lines only covers a small portion of breast cancer cell heterogeneity. Therefore, it is essential to investigate more cell lines using the proposed framework of this study to ensure the generalizability of the findings to a broader range of cell types.

Despite its limitations, our approach offers a mechanistic understanding of the efficacy of drug combinations. Additionally, it provides valuable direction for selecting promising drug pairs worthy of further laboratory investigation.

## Supporting information

**S1 File. Boolean rules of the generic and cell-line specific models.**
(XLSX)

**S1 Table. The missense mutations of five cell lines from cell model passports and DepMap.**
(DOCX)

**S2 Table. Examples of drug pairs with inconsistent synergy scores across five cell lines.**
(DOCX)

**S1 Fig. Clustering analysis of gene expression profiles.**
(DOCX)

**S2 Fig. The frequency of drug target proteins found in false-positive predictions.**
(DOCX)

## Acknowledgments

The authors acknowledge computational resources from the NSTDA Supercomputer Center (ThaiSC). The authors appreciate the editor and reviewers' valuable comments and suggestions, which have greatly improved the manuscript.

## Author Contributions

**Conceptualization:** Marasri Ruengjitchatchawalya, Monrudee Liangruksa, Teeraphan Laomettachit.

**Formal analysis:** Kittisak Taoma, Marasri Ruengjitchatchawalya, Monrudee Liangruksa, Teeraphan Laomettachit.

**Funding acquisition:** Monrudee Liangruksa, Teeraphan Laomettachit.

**Investigation:** Kittisak Taoma, Teeraphan Laomettachit.

**Methodology:** Kittisak Taoma, Marasri Ruengjitchatchawalya, Monrudee Liangruksa, Teeraphan Laomettachit.

**Supervision:** Teeraphan Laomettachit.

**Visualization:** Kittisak Taoma.

**Writing – original draft:** Kittisak Taoma, Teeraphan Laomettachit.

**Writing – review & editing:** Kittisak Taoma, Marasri Ruengjitchatchawalya, Monrudee Liangruksa, Teeraphan Laomettachit.

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
