## [Decision Letter · Decision Letter 0]

9 Nov 2023

PONE-D-23-31129Boolean modeling of breast cancer signaling pathways uncovers mechanisms of drug synergyPLOS ONE

Dear Dr. Laomettachit,

Thank you for submitting your manuscript to PLOS ONE. After careful consideration, we feel that it has merit but does not fully meet PLOS ONE’s publication criteria as it currently stands. Therefore, we invite you to submit a revised version of the manuscript that addresses the points raised during the review process.

We look forward to receiving your revised manuscript.

Kind regards,

Aniruddha Datta

Academic Editor

PLOS ONE

Journal Requirements:

Reviewers' comments:

Reviewer's Responses to Questions

**Comments to the Author**

1. Is the manuscript technically sound, and do the data support the conclusions?

Reviewer #1: No

Reviewer #2: Partly

Reviewer #3: Yes

2. Has the statistical analysis been performed appropriately and rigorously? 

Reviewer #1: I Don't Know

Reviewer #2: N/A

Reviewer #3: Yes

3. Have the authors made all data underlying the findings in their manuscript fully available?

Reviewer #1: Yes

Reviewer #2: Yes

Reviewer #3: Yes

4. Is the manuscript presented in an intelligible fashion and written in standard English?

Reviewer #1: Yes

Reviewer #2: Yes

Reviewer #3: Yes

5. Review Comments to the Author

Reviewer #1: The manuscript requires significant revisions; the authors must explain in detail how the Boolean models were constructed for each cell type and, what is being measured by the model, what the key takeaway is. From reading the manuscript at this stage, it needs to be clarified what is being measured or validated. The Drug synergy score, which is a pivotal part of this work, needs a more detailed explanation as well.

The work, in general, has a lot of potential; there is the modeling of BC pathways, there is drug dosage consideration, and the model takes into account the dynamic behavior of the biological system. I encourage the authors to submit a significantly revised manuscript with more clarity. Following are my comments where the authors can improve their current work.

Page 2, Introduction: Why did the authors limit themselves to two drugs or drug pairs ? Is that usually the maximum number of targeted therapy drugs administered to patients? Please cite the reason along with the appropriate reference.

Page 3, Introduction: Plos One has a readership of diverse backgrounds, so it will be appropriate for authors to explain drug synergy in detail. Authors should explain how drug synergy is measured in their domain of work. This is also not clarified later in the methods section.

Page 3, Introduction, Line 65: Please explain the reason why a Boolean framework was selected.

Page 3, Methods , Line 76-79: Why these cell lines and why 3 TNBC cell lines and 2 ER positive ? What impact does the unequal number of cell lines have on your findings ?

Page 4, Network Reconstruction, Line 96-99: Authors should clearly state if they chose all the pathways for breast cancer from Kegg. Furthermore, which proteins and interactions were picked from SIGNOR2.0 and SIgnaLink , should be mentioned in the manuscript text. Also, why did the authors not select pathways specific to ER+ and TNBC ?

Page 4, Network Reconstruction Line 99: When the authors mention network, they should clarify what sort of network is it (Flow charts, boolean network etc) .

Page 5, Generic Boolean Model, Line 113: Authors should explain in detail why a boolean modeling approach is appropriate for modeling protein activity.

Page 5, Generic Boolean Model, Line 123: Why were inhibitory regulators chosen to be modeled using an AND gate instead of an OR gate ? What is the biological significance.

Page 5, Cell Line Specific Model, Line 123: The authors need to be more specific about tailoring cell line-specific Boolean models. Did they add/remove pathways based on ER+ or TNBC? If so, which pathways were considered and which were removed from the original network?

Page 6, Cell Line Specific Model, Line 144: It is unclear what the Boolean model is measuring and what is being validated. The authors need to expand their model explanation in the previous and current section.

Page 7 , Cell Line Specific Model, Line 162: Can the authors provide an example from their simulation of these incosistencies ?

Page 7, Drug Combination Simulations, Line 172: How did the authors normalize standard dosage information to 0, 0.25,0.75 and 1 ?

Why only these four discrete probablities instead of a continous distribution ?

Page 8, Drug Combination Simulations, Line 185: The authors should clearly define the meaning of PSS and its range of value.

Reviewer #2: The authors are trying to tackle an important problem of find an ideal drug combination using synergistic scores.

1) The authors say that their approach shares similarity with Tsirvouli et al. (2020) and that they have extended it for 5 cell lines. It will be nice to know what other distinctions and improvements the study has made.

2) It will be easy for reader if the authors can show the Boolean model equivalent for Fig. 2.

3) I feel details are missing on how to recreate the models the authors are proposing. This will make it challenging for others to replicate and build upon their studies.

4) The authors say it is a Boolean model but then they refer to normalized/scaled expression values between 0 and 1. They should clarify this in the text.

5) They use asynchronous updates for their model. I believe they should look into literature and have an idea of time-scales of the different reactions and apply update times accordingly. They can incorporate a delay counter into their model.

6) I am not sure how acceptable is the idea that 0.25 of a drug has a 25% chance of inhibiting the protein.

7) There is no explanation for Fig. 3 and how it correlates with the published literature. There is no interpretation of the output by the authors.

I believe the authors have to rework the manuscript to make it easily readable as well as explain their algorithms and figures better.

Reviewer #3: Please find my comments in the attached PDF file.

6. PLOS authors have the option to publish the peer review history of their article (what does this mean?). If published, this will include your full peer review and any attached files.

Reviewer #1: No

Reviewer #2: No

Reviewer #3: No

---

## [Author Response · Author response to Decision Letter 0]

13 Jan 2024

We would like to thank the editor and the reviewers for taking the time to provide feedback on our work. We have carefully considered your suggestions and made the necessary changes to the manuscript. The details of the revisions can be found in our Response to Reviewers.

---

## [Decision Letter · Decision Letter 1]

31 Jan 2024

Boolean modeling of breast cancer signaling pathways uncovers mechanisms of drug synergy

PONE-D-23-31129R1

Dear Dr. Laomettachit,

We’re pleased to inform you that your manuscript has been judged scientifically suitable for publication and will be formally accepted for publication once it meets all outstanding technical requirements.

Kind regards,

Aniruddha Datta

Academic Editor

PLOS ONE

Additional Editor Comments (optional):

Reviewers' comments:

Reviewer's Responses to Questions

**Comments to the Author**

1. If the authors have adequately addressed your comments raised in a previous round of review and you feel that this manuscript is now acceptable for publication, you may indicate that here to bypass the “Comments to the Author” section, enter your conflict of interest statement in the “Confidential to Editor” section, and submit your "Accept" recommendation.

Reviewer #1: All comments have been addressed

Reviewer #2: All comments have been addressed

Reviewer #3: All comments have been addressed

2. Is the manuscript technically sound, and do the data support the conclusions?

Reviewer #1: Yes

Reviewer #2: Yes

Reviewer #3: Yes

3. Has the statistical analysis been performed appropriately and rigorously? 

Reviewer #1: Yes

Reviewer #2: Yes

Reviewer #3: Yes

4. Have the authors made all data underlying the findings in their manuscript fully available?

Reviewer #1: Yes

Reviewer #2: Yes

Reviewer #3: Yes

5. Is the manuscript presented in an intelligible fashion and written in standard English?

Reviewer #1: Yes

Reviewer #2: Yes

Reviewer #3: Yes

6. Review Comments to the Author

Reviewer #1: I commend the authors for taking the time to address my concerns and answer my questions. Their work has significantly improved the article and will be a great addition to computational research of drug effects in breast cancer. I wish them good luck with the rest of submission.

Reviewer #2: (No Response)

Reviewer #3: (No Response)

7. PLOS authors have the option to publish the peer review history of their article (what does this mean?). If published, this will include your full peer review and any attached files.

Reviewer #1: No

Reviewer #2: No

Reviewer #3: No

---

## [Editor Report · Acceptance letter]

13 Feb 2024

PONE-D-23-31129R1 

PLOS ONE

Dear Dr. Laomettachit, 

I'm pleased to inform you that your manuscript has been deemed suitable for publication in PLOS ONE. Congratulations! Your manuscript is now being handed over to our production team.

Kind regards, 

on behalf of

Dr. Aniruddha Datta 

Academic Editor

PLOS ONE